# Adaptive microwave impedance memory effect in a ferromagnetic insulator

Hanju Lee[1], Barry Friedman[2] & Kiejin Lee[1]

Adaptive electronics, which are often referred to as memristive systems as they often rely on a memristor (memory resistor), are an emerging technology inspired by adaptive biological systems. Dissipative systems may provide a proper platform to implement an adaptive system due to its inherent adaptive property that parameters describing the system are optimized to maximize the entropy production for a given environment. Here, we report that a non-volatile and reversible adaptive microwave impedance memory device can be realized through the adaptive property of the dissipative structure of the driven ferromagnetic system. Like the memristive device, the microwave impedance of the device is modulated as a function of excitation microwave passing through the device. This kind of new device may not only helpful to implement adaptive information processing technologies, but also may be useful to investigate and understand the underlying mechanism of spontaneous formation of complex and ordered structures.

[1] Department of Physics and Basic Science Institute for Cell Damage Control, Sogang University, Seoul 121-742, Korea. [2] Department of Physics, Sam Houston State University, Huntsville, Texas 77341, USA. Correspondence and requests for materials should be addressed to K.L. (email: klee@sogang.ac.kr).

Dissipative systems that are driven far from equilibrium adapt to changes of the thermodynamic environment by modifying their internal structure to maximize the system's entropy production according to the maximum entropy production principle[1] (MEPP). Such systems often yield a counterintuitive result that a complex and highly ordered structure, which is called the dissipative structure, appears spontaneously from a disordered and strongly fluctuating state.

It is interesting to consider a device that operates based on the dissipative structure. This device may modulate its internal structure in response to a given thermodynamic environment to maximize the system's entropy production, and may modify even the environment through an improved interaction between the internal structure of the system and environment. From the electronics point of view, it behaves the same as memristive circuit elements[2–11]: the electric conductivity (internal state) is modulated as a function of applied electric voltage or current (environment), and the modulated conductivity changes the applied electric voltage or current in return. In particular, if the device has a non-volatile memory effect, it would satisfy basic requirements of an element required for an adaptive electronics and information processing system: a multiply-valued and history-dependent internal state that can be tuned in a reversible and non-volatile manner[2,3].

In this report, we show that the non-volatile memristive microwave device can be realized from the adaptive property of the driven ferromagnetic system. Unlike the conventional memristive devices that are controlled by a static (or a slowly varying) electromagnetic field, the microwave response of the present device is controlled by the frequency, amplitude and incident direction of the applied microwave signal. This full-wave operation is resulted from the adaptive tuning of the magnetic domain (MD) structure of the ferromagnetic material for the driving microwave signal to maximize the energy dissipation of the system.

## Results

**Operation principle**. Figure 1a–b shows an illustration of the present device, where the yttrium iron garnet (YIG) single crystalline thin film having an in-plane magnetic anisotropy interacts with a strong microwave magnetic field of the high impedance line (HIL) of the stepped impedance low pass filter (SILPF). The operation principle of present device is identical to that of the tunable YIG-based microwave devices that device response is modulated by resonance frequency tuning of the coupled YIG material[12,13] (Fig. 1c–f). However, the present device operates based on the natural ferromagnetic resonance (N-FMR; also called the zero-field resonance) effect[14–16], and the N-FMR frequency is tuned by modifying the MD structure of the YIG through an intense microwave signal, where the YIG material has a particular importance due to its outstanding properties such as a high quality factor (Q-factor) with a narrow FMR line-width and extremely low loss[17].

Figure 1g illustrates the operation principle of the N-FMR device proposed in the present study. Under no (or a weak) external static magnetic field, ferromagnetic materials form MDs to minimize their magnetic free energy (initial state). When a microwave signal is applied to the device, the magnetic moments of MDs start to precess around the axis of their magnetic moments (Fig. 1d), where the precession amplitude depends on the frequency and power of the applied microwave signal and the N-FMR frequency of the MDs. When a weak microwave signal is applied to the MDs so that the precession amplitude is small, the precession energy is not enough to modify their internal effective field. Then, the response of the device will be not changed by the precession, and the response depends on the N-FMR frequency of the initial MD structure (probing process). However, when an intense microwave is applied, the strong precession energy can change the MD structure[18–21]. If the modified MD structure is stable so that there is no internal torque acting on its magnetic moment, it will remain even when the microwave signal is stopped (modulation process). Then, the device response will show a non-volatile change from a variation of the N-FMR frequency of the modified MD structure.

This non-volatile response change can be described by using the scattering parameter[22] (S-parameter) with the microwave memory impedance (microwave mempedance), where we used the term microwave mempedance due to its similarity to the memristor[10,11] (memory resistor) that the impedance of the device changes as a function of the past microwave signal. As the response of present device is governed by the N-FMR effect of the YIG, one can describe the microwave mempedance as a non-volatile change of the magnetic susceptibility of the YIG. Then, for a weak microwave signal, one can find the relation between the changes of transmission response of the device and the magnetic susceptibility of the YIG as[23–26] (Supplementary Note 1):

$$\Delta A \sim \Delta \mathrm{Re}\{\log S_{21}\} \sim -\Delta \chi'', \qquad (1)$$

$$\Delta \varphi \sim -\Delta \mathrm{Im}\{\log S_{21}\} \sim -\Delta \chi', \qquad (2)$$

where, $\Delta A$ and $\Delta \varphi$ are the changes of the amplitude and phase of the transmitted microwave, $S_{21}$ describes the transmittance response at port 2 due to a signal at port 1, and $\chi'$ and $\chi''$ are the real and imaginary parts of the magnetic susceptibility of the YIG. equations (1) and (2) indicate that the amplitude and phase of transmitted microwave changes as a function of imaginary and real parts of the magnetic susceptibility of the YIG, and that the changes can be confirmed through the $S_{21}$-parameter measurements.

For a weak microwave signal, the magnetic susceptibility is a function of frequency, and shows a strong change around the resonance condition. Therefore, the response change of the device is maximized around the N-FMR frequency, where the N-FMR frequency is determined by the demagnetization field energy of the MD structure. For a simple case that only the N-FMR frequency is varied by the MD modulation process, one can express the change of device response as[20,27] (Supplementary Note 1):

$$\Delta A \sim \frac{\Delta F_{xy}}{F_{xy}} s(1-s) \sim \frac{\Delta \omega_{\mathrm{r}}}{\omega_{\mathrm{r}}} s(1-s), \qquad (3)$$

$$\Delta \phi \sim \frac{\Delta F_{xy}}{F_{xy}} s \sim \frac{\Delta \omega_{\mathrm{r}}}{\omega_{\mathrm{r}}} s, \qquad (4)$$

$$F_{xy} = F_x - F_y, \quad s = \frac{\omega}{\omega_{\mathrm{r}}}, \qquad (5)$$

where, $\omega_{\mathrm{r}}$ is the N-FMR frequency of the initial MD state, $F_x$ and $F_y$ are the demagnetization field energies for the two Cartesian axis along the film plane, $\omega$ is the weak microwave frequency. The demagnetization field energy is a function of demagnetization factor of the MDs, and the demagnetization factor depends on the geometrical shape of MDs[20,28]. Because the MD structure is determined by a magnetic history during the magnetization process, the device response will depend on the history of modulation process by an intense microwave signal, and thus, the device will show memristive behavior for the input microwave signal.

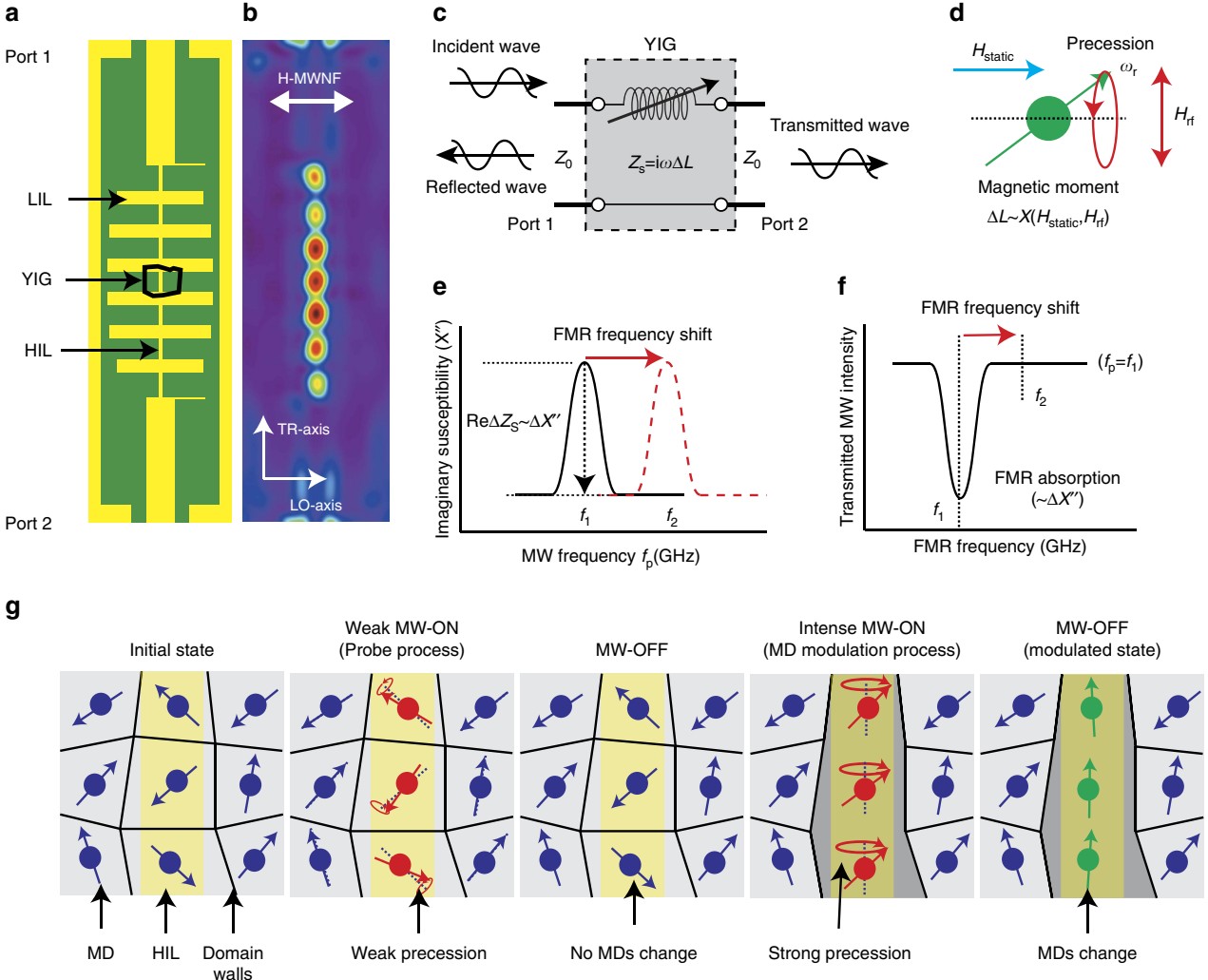

**Figure 1 | Illustration of the operation principle of the N-FMR device.** (**a,b**) Illustration of the experimental setup (**a**) and the magnetic microwave near field structure (H-MWNF; **b**) simulated at 1.0 GHz (COMSOL Multiphysics). The stepped impedance low pass filter (SILPF) consists of a cascade of low impedance lines (LIL) alternately connected with high impedance lines (HIL), where the H-MWNF distribution of the SILPF is localized around the HILs. The yttrium iron garnet (YIG) thin film grown on GGG (111) substrate is placed on the HIL of the SILPF, and thus, it interacts with the H-MWNF of the SILPF. Two axes are defined according to the magneto optical Kerr effects (MOKE): longitudinal and transverse MOKE effect for the LO-axis and the TR-axis, respectively, and the polarization direction of the H-MWNF is parallel to the LO-axis. (**c**) Equivalent circuit model for the ferromagnetic resonance (FMR) device, where the impedance change ($Z_s$) of the device by the YIG is described by a variable inductor ($\Delta L$) connected in series with the waveguide system ($Z_0$). (**d**) Illustration of the FMR effect: the magnetic moment under a static magnetic field ($H_{static}$) precess by the microwave magnetic field ($H_{rf}$), where the polarization directions of the $H_{static}$ and $H_{rf}$ are perpendicular to each other. The change of inductance of the device is a function of magnetic susceptibility ($\chi$) of the YIG for the $H_{rf}$, where the $\chi$ is determined by the FMR frequency ($\omega_r$) that depends on the $H_{static}$. (**e,f**) Illustrations for the change of microwave transmittance response by the FMR frequency shift. When the FMR frequency increases from $f_1$ to $f_2$, the imaginary part of the $\chi$ ($\chi''$) at $f_1$ is decreased by the FMR frequency shift, and as a result, the transmitted microwave intensity at $f_1$ is increased by a decrease of FMR absorption that is a function of $\chi''$. (**g**) Illustrations on the operation principle of the N-FMR device for the probing and the modulation process.

**Microwave response modulation from arbitrary initial states.** To verify the response modulation by the N-FMR effect, we measured the $S_{21}$-parameter changes of the present device caused by an intense microwave signal. Here, we use the term probing microwave (frequency: $f_p$; power: $p_p = 0$ dBm) for the weak microwave, and use the term modulation microwave (frequency: $f_m$; power: $p_m$) for the intense microwave. We calculated differences of the $S_{21}$-parameter of the probing microwave ($\Delta S_{21,p}$) between the initial and modulated state, where the initial states of the device were refreshed before each modulation processes by applying a static magnetic field for a second. Therefore, the measurement result can describe overall behavior of the device response modulation from arbitrary initial states.

Figure 2a–d shows averaged variations of the $\Delta S_{21,p}$ as a function of $f_m$ and $p_m$. From the $f_m$-sweep measurement results (Fig. 2a,b), one can see that there is a frequency region (0.5–1.5 GHz) showing a strong change of the $\Delta S_{21,p}$. In this frequency range, the real and imaginary parts of the $\Delta S_{21,p}$ showed a typical spectral line-shapes of the magnetic susceptibility appearing in the vicinity of the magnetic resonance frequency (Fig. 2e): the real part of the magnetic susceptibility (imaginary part of the $\Delta S_{21,p}$; Im-$\Delta S_{21,p}$) shows an anomalous dispersion, while the imaginary part (real part of the $\Delta S_{21,p}$; Re-$\Delta S_{21,p}$) shows a resonant absorption (Lorentzian or Gaussian) line-shapes. Figure 2f shows calculated N-FMR frequencies by fitting vertical line-profiles of the $\Delta S_{21,p}$ to the Gaussian

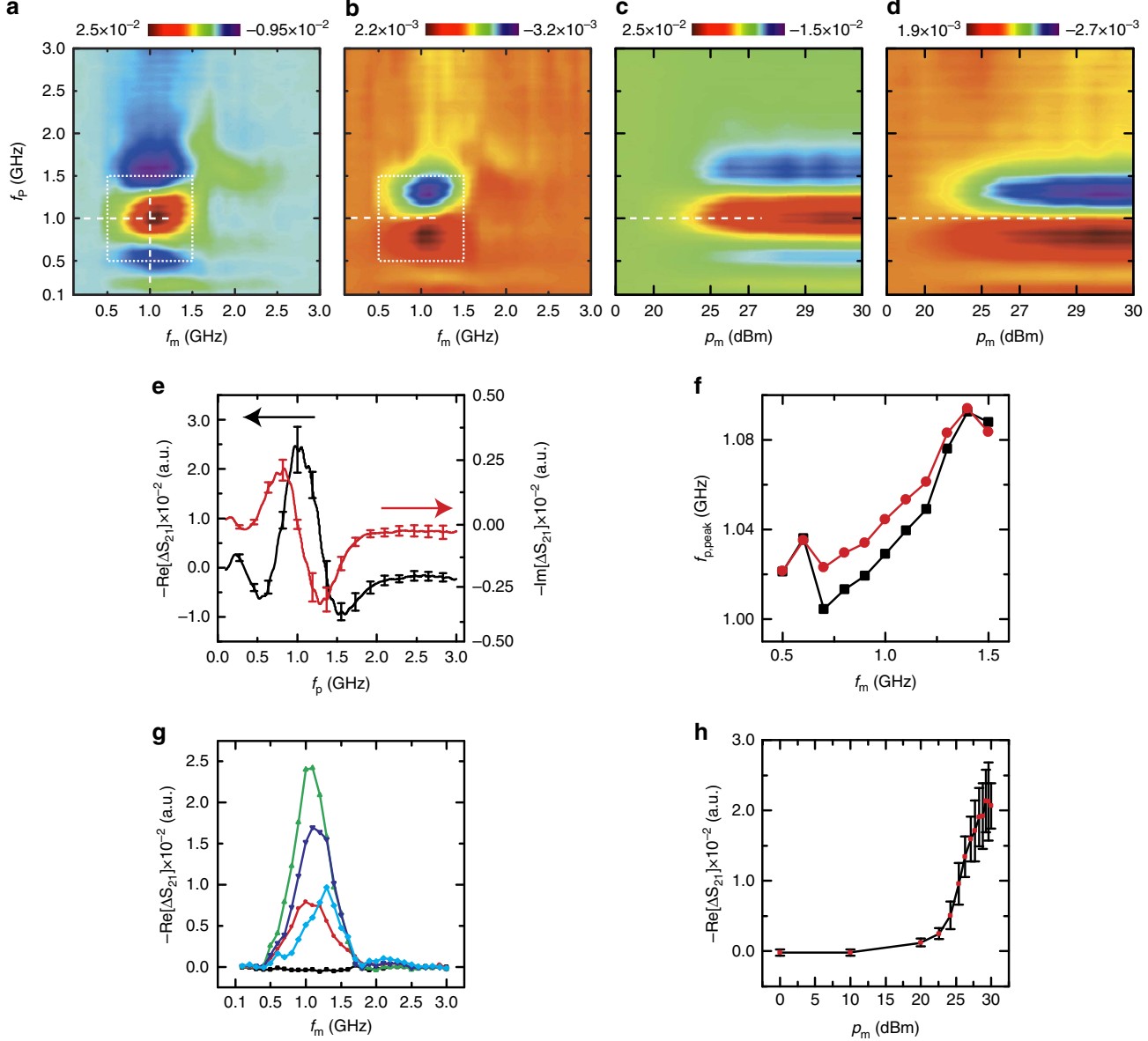

**Figure 2 | Microwave response changes by the modulation microwave for arbitrary initial states.** (**a–d**) Differences of the $S_{21}$-parameter of the probing microwave ($\Delta S_{21,p}$) between the initial and modulated state as a function of modulation microwave frequency ($f_m$; **a**: real part, **b**: imaginary part) and power ($p_m$; **c**: real part, **d**: imaginary part), where subscripts p and m denote the probing and modulation microwaves, respectively. (**e**) Line-profiles at $f_m = 1.0$ GHz for the real (black) and imaginary (red) parts of the $\Delta S_{21,p}$. The error bars represent the s.d. of the averaged (20-times) data. (**f**) Calculated N-FMR frequencies from real (black; Re[$\Delta S_{21,p}$]) and imaginary (red; Im[$\Delta S_{21,p}$]) $\Delta S_{21,p}$ line profiles as a function of $f_m$. (**g**) Changes of the Re[$\Delta S_{21,p}$] as a function of $f_m$ for various $f_p$ (black: 0.1 GHz; red: 0.8 GHz; green: 1.0 GHz; blue: 1.2 GHz; cyan: 1.3 GHz). (**h**) Changes of the Re[$\Delta S_{21,p}$] as a function of modulation microwave power ($p_m$) at $f_p = 1.0$ GHz and $f_m = 1.0$ GHz. The error bars represent the s.d. of the averaged (20-times) data.

line-shape functions[23,29] (for details, see Methods). The calculation result showed that the N-FMR frequency increases as the $f_m$ increases, where the variation was limited in the range from 1.0 to 1.1 GHz. This result is consistent with the fact that the N-FMR frequency of the polycrystalline YIG material is typically around 1.0 GHz (ref. 16), and indicates that the modulation microwave enhances the N-FMR effect of the YIG.

Line profiles along the horizontal direction of the $\Delta S_{21,p}$ around $f_p = 1.0$ GHz shows an another interesting character that the $\Delta S_{21,p}$ peak occurs at a higher $f_m$ as the $f_p$ increases. Figure 2g shows the line profiles of real part of the $\Delta S_{21,p}$ for various $f_p$, where one can see that the peaks for $f_p = 0.8$–1.0 GHz occur when the $f_m$ is around 1.0 GHz, while the peak for $f_p = 1.3$ GHz occurs when the $f_m$ is around 1.3 GHz. These line profiles correspond to

the microwave absorption change of the device, and thus, this result can be interpreted as that a higher $f_m$ results in an increase of the microwave absorption at a higher frequency region by increasing the N-FMR frequency. In addition, there was no significant spectral change of the $\Delta S_{21,p}$ depending on the $p_m$ as shown in Fig. 2c,d, and this indicates the N-FMR frequency shift is only dependent on the $f_m$ in present power levels (0–30 dBm). The change of Re-$\Delta S_{21,p}$ occurred when the $p_m$ exceeded 20 dBm (Fig. 2h), and this indicates the threshold behavior of the modulation that it occurs when the precession amplitude exceeds some critical value.

The enhanced N-FMR effect indicates that there is a change of the MD structure in the YIG. Figure 3a–b shows optical image of the YIG thin film and illustrations of the experimental setup and

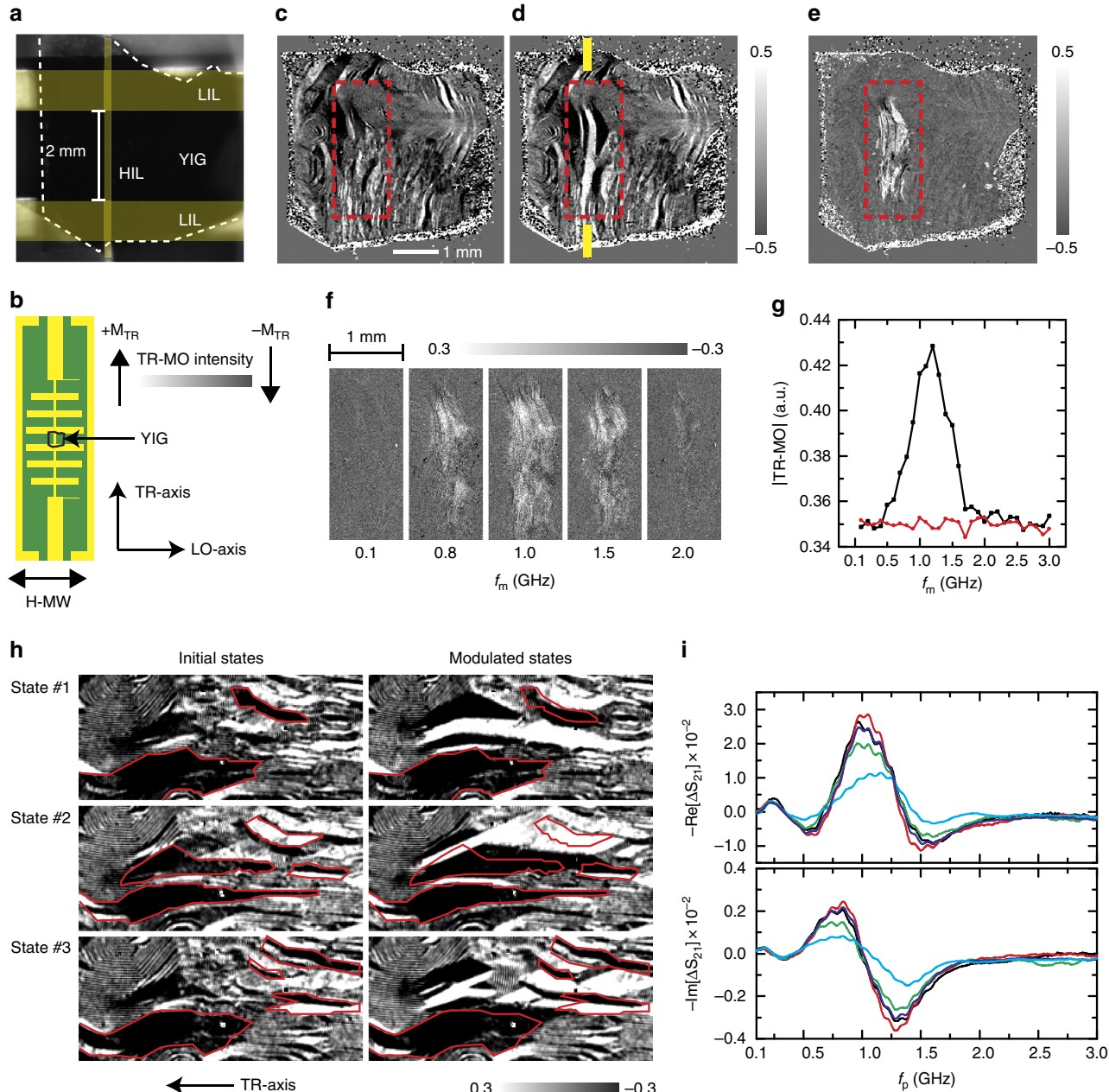

**Figure 3 | Variation of magnetic domain structure by the modulation microwave from arbitrary initial states.** (**a**) Optical image of the YIG thin film (indicated by white dashed line). (**b**) Illustrations of the experimental setup and axes, where the polarization direction of the H-MWNF is parallel to the LO-axis, and the positive and negative transverse MOKE signal (TR-MO) correspond to magnetic moments whose polarization directions are parallel and anti-parallel to the TR-axis, respectively. (**c,d**) Magnetic domain images measured at the initial (**c**) and the modulated (**d**) states, where the $f_m = 1.0$ GHz. (**e**) Difference of absolute TR-MO ($\Delta$|TR-MO|) intensity between the initial and the modulated states. (**f**) Averaged $\Delta$|TR-MO| distribution for various $f_m$, where the positive sign of $\Delta$|TR-MO| correspond to a population of magnetic moment whose polarization direction is perpendicular to the LO-axis. (**g**) Averaged |TR-MO| intensities measured at the initial (red) and modulated (black) states as a function of $f_m$, where the measurements were conducted 20 times with refreshing the initial states for each $f_m$. (**h**) Detailed MD structures of the initial and the modulated states at $f_m = 1.0$ GHz, where the traces of initial MDs that remain after the modulation are indicated by red lines. (**i**) Real and imaginary parts of $\Delta S_{21,p}$ corresponding to the MD structures (black: state #1; red: state #2; green: state #3). In addition, averaged $\Delta S_{21,p}$ at $f_m = 1.0$ GHz (blue) and $f_m = 1.4$ GHz (cyan) are also presented for the comparison.

axes, and Fig. 3c-d shows representative measurement results on the modification of the MD structure, where the MDs were visualized by magneto-optical (MO) microscope system based on the transverse Kerr MO effect[30,31] (for details, see Methods). From the measurement results, one can see that MDs whose magnetic moments are parallel (bright area) or anti-parallel (dark area) to the TR-axis are populated around the HIL. The change of

MD structure locally occurred around the HIL for all $f_m$ as shown in Fig. 3e,f, and it occurred strongly when the $f_m$ is in the range from 0.5 to 1.5 GHz as shown in Fig. 3g. Figure 3h shows detailed MD structures of the initial and the modulated states at $f_m = 1.0$ GHz, where the state #1 showed $\Delta S_{21,p}$ close to the average value, and state #2 and #3 showed deviations of the $\Delta S_{21,p}$ from the average value in positive and in negative, respectively.

From the MD images, one can see that while the growth of the MDs occurs for all states, their local structures are different to each other. As shown in Fig. 3i, the $\Delta S_{21,p}$ of each states showed deviations in their amplitudes, however, their line shapes were close to each other and were clearly different to that of modulated state at a higher $f_m$ (1.4 GHz). Because the $\Delta S_{21,p}$ line-shape is a function of N-FMR frequency, this result indicates that the N-FMR frequencies are close to each other if the modulation microwave frequencies are identical, although the modulated MD structures are different from each other

The growth of the MDs occurred in common for all $f_m$ showing $\Delta S_{21,p}$ changes, however, there was no apparent character of the MD structures depending on the $f_m$. In addition, the MDs showed a dependency on their initial structures. We indicated the traces of initial structures that remained after the modulation process in Fig. 3h. By comparing the initial and modulated MDs, one can find that there is a growth (or a reduction) of domain width around the traces of the initial MD structure. The traces appeared more clearly when they were far from the HIL, while the appearance of the traces was somewhat arbitrary near the HIL, where some initial MDs disappeared with creations of the new MD structures. This indicates that the growth of the MD occurs in a stochastic way, and the initial MD structure plays an important role on the domain growth.

Regarding the growth of the MD, a similar result has been reported for a Landau flux-closure structure[20]. The authors showed that when an intense microwave with a slightly off resonance frequency was applied, the MDs forming the flux-closure structure turned into a single domain state through a domain wall shift, where the magnetization of the single domain was perpendicular to the driving microwave magnetic field. About this behavior, the authors proposed that the MEPP is responsible: the domain wall moves to increase the total precession energy so that the entropy production of the system is maximized. The enhancement of the N-FMR effect and the shift of N-FMR frequency observed in the $\Delta S_{21,p}$ measurement can be explained in the same way. The growth of MDs whose magnetization is perpendicular to the polarization direction of the microwave magnetic field is beneficial to increase the precession energy. In this perpendicular configuration, the MDs can absorb the incident microwave more efficiently because the torque acting on the magnetic moment is maximized, and thus, the N-FMR effect is enhanced. In particular, because the entropy production of driven MDs increases as their precession energy increases, they will evolve to have a resonance frequency close to the modulation microwave frequency according to the MEPP. This well describes the $\Delta S_{21,p}$ measurement results that the modulation microwave results in an enhancement of the N-FMR effect, and the increase of N-FMR frequency along with the increase of modulation microwave frequency.

**Reversible microwave response modulation**. To verify the reversibility of the device response modulation, we conducted $\Delta S_{21,p}$ measurements with continuous sweeps of the $f_m$ and $p_m$ for a given initial MD structure. A series of experiments were conducted with various initial MD structures, and there were no fundamental differences between the measurement results. Here, we focused on the $f_m$-sweep results, and descriptions on the $p_m$-sweep are presented in the Supplementary Note 2. Figure 4a–d shows representative measurement results of the $\Delta S_{21,p}$ observed in the initial and after the initial $f_m$-sweep, and Fig. 4e,f shows the $\Delta S_{21,p}$ changes as a function of sweep index. From the measurement results, one can see clearly that the $\Delta S_{21,p}$ change occurred reversibly after the initial forward $f_m$-sweep. The $\Delta S_{21,p}$ change occurred strongly when the $f_m$ was in the range from 1.0

to 1.5 GHz, otherwise it showed a small variation when the $f_m$ is out of this frequency range. This can be explained as that the precession amplitude rapidly decreased as the $f_m$ got away from the N-FMR frequency ($\sim 1$ GHz), and the weak precession cannot modify the modulated MD structure during the initial sweep. We note that because the MD modulation is driven by the precession energy, a more intense modulation microwave will modify the MD structure even if its frequency is far from the N-FMR frequency. This comes from the fact that the precession amplitude is not only a function of driving field frequency but also a function of driving field strength.

Figure 4g,h shows line line-profiles of real and imaginary parts of $\Delta S_{21,p}$ for various $f_m$, and Fig. 4i shows the N-FMR frequencies calculated from the line line-profiles. All the line profiles showed a line shape of a magnetic permeability occurring near the resonance, and their N-FMR frequency increases as the $f_m$ increases. As we indicated in Fig. 4g, the increase of N-FMR frequency resulted in an increase of microwave absorption at a higher frequency region. Figure 4j shows changes of the real part of the $\Delta S_{21,p}$ for $f_p = 1.0$, 1.2, and 1.4 GHz as a function of $f_m$, where one can see that the $\Delta S_{21,p}$ are maximized when the $f_m$ and $f_p$ are close to each other. This result is consistent with the MEPP as discussed above, where the MD structure is modulated to have the N-FMR frequency close to $f_m$ so that the energy dissipation is maximized, and as a result, the microwave absorption around $f_m$ is increased through the N-FMR frequency shift.

Figure 5a–c shows the changes of the MD structure occurring in the initial $f_m$-sweep (a) and in the first (sweep #1; b) and the final (sweep #10; c) sweeps after the initial $f_m$-sweep. During the initial $f_m$-sweep, the MDs grew irreversibly through a domain wall shift of the pre-existing MDs, where some MDs were created during the $f_m$-sweep, and they acted as a pre-existing MD for subsequent $f_m$-sweep. The growth of the MD was finished at $f_m = 1.0$ GHz, and the grown MDs remained and showed a reversible change during subsequent modulations. This result indicates that the grown MDs during the initial modulation are stable to subsequent modulation processes, and the change of MDs is restricted by the stable grown MDs. Three kinds of reversible changes were observed: (A1) a shift of domain position; (A2) a local change of magnetic orientation; (A3) a complicated structural change. The A1 domain was shifted from its original place along the LO-axis as the $f_m$ increased as shown in Fig. 5d,e, and returned again during the backward $f_m$-sweep. At the same time, a local MO intensity of A2 nearby A1 slightly increased showing an inhomogeneous spatial distribution. The increase of MO intensity in the A2 indicates a decrease of the magnetic moments oriented along the TR-axis, and the inhomogeneous MO signal indicates spatial fluctuations of the magnetic moments. The MD changes in A1 and A2 occurred simultaneously, and this result indicates that the changes of MDs in A1 and A2 are coupled to each other. Otherwise, there was a complicated change of MD structure in A3, where the MO signal showed an inhomogeneous spatial distribution with no distinct domain wall. Figure 5f shows averaged absolute value of spatial gradient of MO images for $f_m = 0.1$ and 3.0 GHz, and Fig. 5g shows its local average changes as a function of $f_m$. From the results, one can see that that the local structural fluctuation of magnetic moment changes reversibly during the $f_m$-sweeps at 1.0–1.5 GHz, and thus, it can be concluded that the complicated MD changes in A3 are also related to the reversible N-FMR frequency changes.

A more detailed study will be required for quantitative explanations on the reversible change of MDs and on the relation between the N-FMR frequency and the modulated MD structure, where it should be based on analysis on the demagnetization field and the magneto-dynamics of coupled MDs composing the

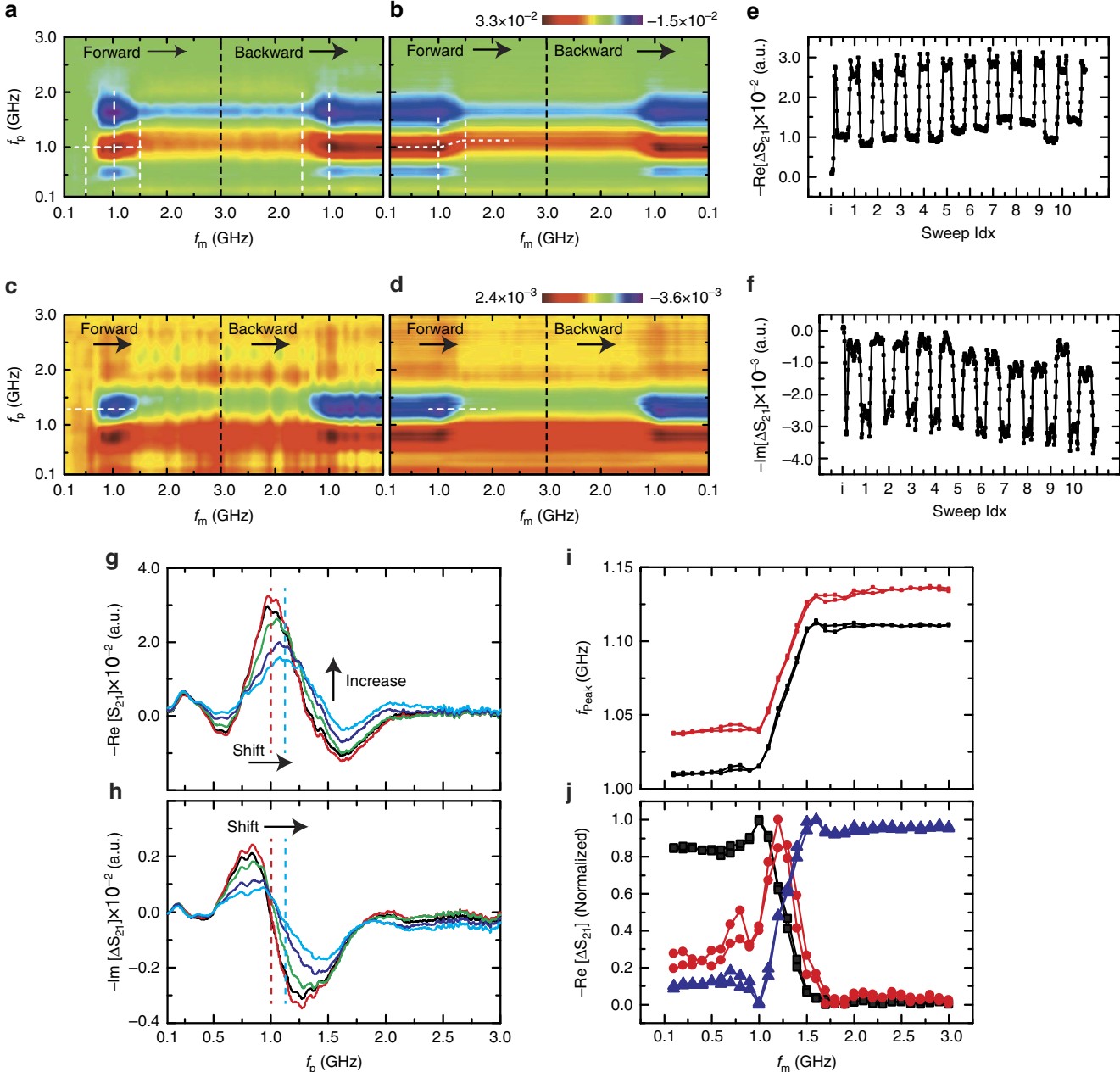

**Figure 4 | Microwave response changes by the $f_m$-sweep for a given initial state.** (a–d) The changes of the Re[$\Delta S_{21,p}$] (a,b) and Im[$\Delta S_{21,p}$] (c,d) as a function of $f_m$ measured during the initial $f_m$-sweep (a,c) and averaged (10-times; b,d) after the initial $f_m$-sweep, where the $f_m$ was swept from 0.1 to 3.0 GHz (forward) and from 3.0 to 0.1 GHz (backward) continuously without refreshing the initial state. (e,f) The changes of the Re[$\Delta S_{21,p}$] at $f_p = 1.0$ GHz and Im[$\Delta S_{21,p}$] at $f_p = 1.2$ GHz as a function of sweep index. (g,h) Line-profiles of Re[$\Delta S_{21,p}$] and Im[$\Delta S_{21,p}$] for various $f_m$ (black: 0.1 GHz; red: 1.0 GHz; green: 1.2 GHz; blue: 1.4 GHz; cyan: 2.0 GHz). (i) N-FMR frequencies calculated from Re[$\Delta S_{21,p}$] (black) and Im[$\Delta S_{21,p}$] (red) line profiles for each $f_m$. (j) The changes of the Re[$\Delta S_{21,p}$] at $f_p = 1.0$ GHz (black), 1.2 GHz (red), and 1.5 GHz (blue) as a function of $f_m$, where the changes were normalized to their maximum values.

complicated structure. We note that kinds of MD structures can occur with various modulation pathways depending on its initial structure, magnetic property of the material, and so on, and each MD structures may require an individual magneto-dynamic analysis. However, although the modulation processes have various pathways, there is a thermodynamic direction that MDs evolve to maximize the entropy production of the system according to the MEPP. This thermodynamic direction results in a MD structure that can more store the precession energy than the initial one, where the increase of precession energy is achieved through an enhanced N-FMR effect of the modified MDs for the

modulation microwave. Therefore, the modulation microwave results in an enhancement of the N-FMR effect around its frequency, and as a result, the device response shows a reversible change.

**Response modulation by near field structure.** The precession energy of the MDs, which is responsible to the modulation process, is not only a function of frequency but also a function of amplitude of the H-MWNF (magnetic microwave near-field structure). In the microstrip microwave device, the amplitude and

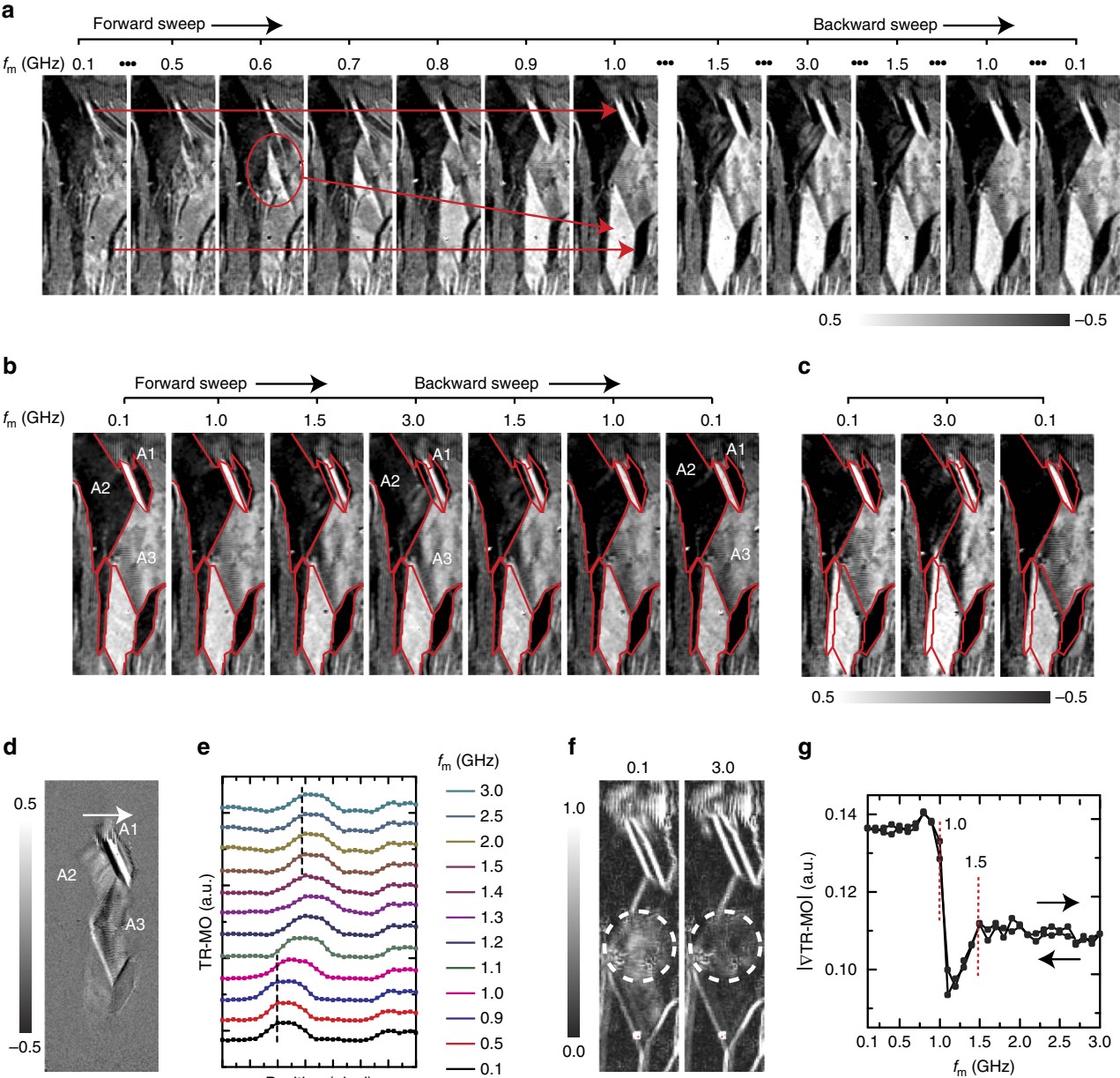

**Figure 5 | MD structure changes by the $f_m$-sweep for a given initial state.** (a) MD images observed during the initial $f_m$-sweep, where initial MDs that remained after the $f_m$-sweep were indicated by red arrows. (b,c) MD images observed during the first (b) and last (10 times; c) $f_m$-sweep, where three MDs that showed distinct reversible changes were indicated by A1–A3. (d) Averaged difference of TR-MO intensity ($\Delta$TR-MO) between the $f_m = 0.1$ and 3.0 GHz, where 20 images were averaged for each $f_m$. (e) The line-profiles of TR-MO intensity from A2 to A1 (indicated by arrow) for various $f_m$. (f) Absolute values of spatial gradient of averaged TR-MO images measured at $f_m = 0.1$ and 3.0 GHz. (g) Local average change of the spatial gradient values around A3 (indicated by dotted circle).

its spatial distribution of the H-MWNF strongly depend on the microwave frequency, and they can be a function of the incident direction of the microwave signal for a nonreciprocal device. Therefore, a more rich and interesting response modulation can be realized by utilizing the spatial distribution of the H-MWNF, where the frequency, power, and incident direction of modulation microwave can be parameters determining the H-MWNF structure. We present practical examples in Supplementary Note 3, where one can see that the device response can be modulated utilizing such parameters.

Figure 6 shows a particular example utilizing the spatial structure of the H-MWNF depending on the microwave frequency for the device response modulation. The measurements were conducted by using a YIG thin film that covers three HILs of

the SILPF, where the film surface was faced to the HILs to enhance the interaction between the MDs and H-MWNF. Details on the measurement techniques for the imaging of MD and H-MWNF and for the microwave transmittance are presented in the Methods section. Figure 6a,b shows averaged changes of transmitted microwave power after the initial $f_m$-sweep, and descriptions for the initial $f_m$-sweep are presented in Supplementary Note 4. From the transmittance changes for forward and backward $f_m$-sweeps, one can see a clear reversible change of the transmittance at $f_P = 1.9$ GHz. The transmitted power at $f_P = 1.9$ GHz decreased (increased) as the $f_m$ approached to 2.0 GHz during the forward (backward) $f_m$-sweep, and showed a small variation during the subsequent sweeps. This result is consistent with the above discussion based on the MEPP that the

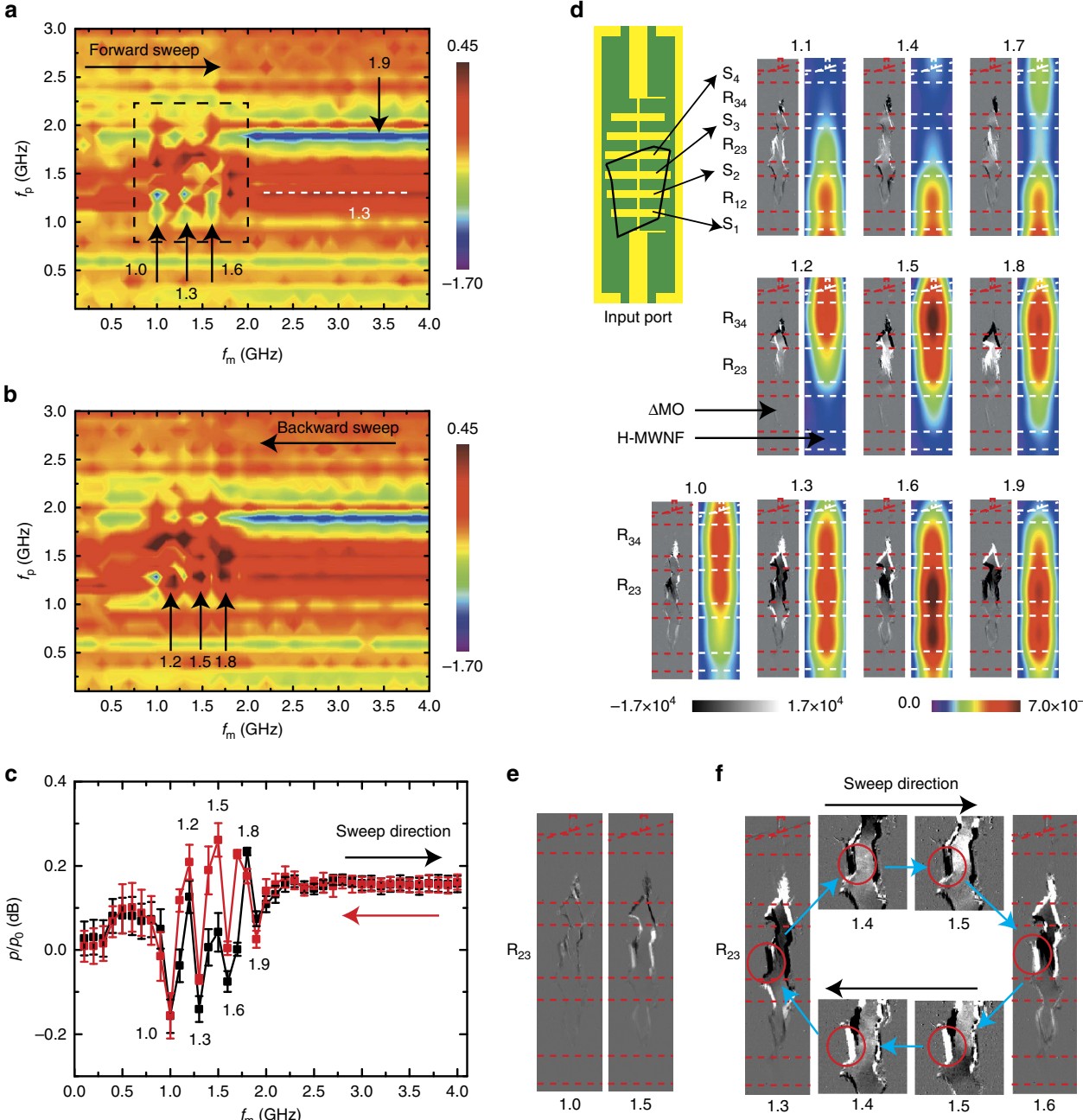

**Figure 6 | Changes of microwave transmittance and MD structure by the H-MWNF structure.** (**a,b**) Transmitted probing microwave power for forward (**a**) and backward (**b**) $f_m$-sweeps, where $p_O$ is the transmitted power measured after initial $f_m$-sweeps, and powers of modulation and probing microwave were 20 and −10 dBm, respectively. The transmitted power was averaged for four modulation sweeps after the initial sweep. Arrows indicate periodic changes of transmitted power. (**c**) Line profile at $f_p = 1.3$ GHz. The error bars represent the s.d. of the averaged data. (**d**) Averaged changes of magnetic domain structure (gray) as a function of $f_m$, and the H-MWNF distribution images (colour). (**e**) Hysteresis of MD modulation at 1.0 and 1.5 GHz, where the hysteresis is a difference of MD images measured in forward and backward sweeps. (**f**) MD modulation processes from 1.3 to 1.6 GHz, where arrows indicate the modulation sweep direction.

modulation microwave results in an increase of microwave absorption around its frequency. In addition, there was a frequency region ($f_p = 1.1$–1.8 GHz) showing a complicated change of the microwave transmittance. Figure 6c shows the microwave transmittance changes at $f_p = 1.3$ GHz as a function of $f_m$, where one can see a periodic change of the microwave transmittance: it decreases when the $f_m$ are 1.0, 1.3, 1.6 and 1.9 GHz; it increases when the $f_m$ are 1.2, 1.5 and 1.8 GHz. From the H-MWNF and MD images shown in Fig. 6d, one can see that a similar H-MWNF structure appears with a period of 0.3 GHz

consistently with the periodic transmittance changes, and the modulated MDs have a similar structure to each other when the H-MWNF distributions are similar to each other. This result indicates that the periodic change of the MDs is responsible for that of the microwave transmittance, and it arises from the MD modulation depending on the spatial distribution of the H-MWNF.

The periodic response change implies that the N-FMR frequency of the modulated MDs is shifted periodically to higher and lower frequency (Supplementary Fig. 11). Although we

discussed that a higher $f_m$ results in an increase of N-FMR frequency and an increase of microwave absorption around the $f_m$, a deviation from such behavior can occur because the N-FMR frequency can be a function of the precession amplitude. It is known that for a high power microwave, the FMR frequency is shifted to a higher or a lower frequency as a function of incident microwave power, where the sign of the shift depends on the magnetic property of the material, applied static magnetic field, excitation microwave frequency, and so on[32–34]. This nonlinear FMR frequency shift may be responsible for the deviation of the N-FMR frequency change. Because the modulated MDs have a structure that maximizes their precession energy to the modulation microwave rather than probing microwave, the N-FMR frequency of the modulated MDs for the probing microwave can ether increase or decrease depending on the nonlinear FMR frequency shift. As shown in Supplementary Note 5 and Supplementary Fig. 9, the decrease of the microwave transmittance commonly occurred when a strong fluctuation of MO signal appeared during the modulation process. The fluctuations occurred when the local H-MWNF amplitude was strong around the fluctuating regions, where the fluctuation of MO signal indicates instability of the MD structure caused by a strong magnetic precession. These results suggest that the nonlinear FMR frequency shift caused by the strong spin precession may be responsible for the periodic N-FMR frequency change. However, a more systematic study will be needed to explain the present result, and we postpone it to our upcoming study. We emphasize that the periodic response change comes from the frequency dependent H-MWNF structure, and thus, the present result demonstrates that the device response can be modulated as a function of H-MWNF structure.

Besides the reversible changes, one can see a large hysteresis of the microwave transmittance around 1.5 GHz from Fig. 6c. Figure 6e shows a hysteresis of the MD structure for excitation frequencies of 1.0 and 1.5 GHz. From the measurement results of MW transmittance and MD hysteresis, one can see that there is a relatively large hysteresis of the transmittance and MD structure for 1.5 GHz compared with 1.0 GHz. Figure 6f shows the MD changes as a function of excitation frequency for forward and backward sweep. From the MD images, one can see that there is a part of MD modulated oppositely at 1.3 and 1.6 GHz (indicated by red circle), and it is unchanged by the excitations at 1.4 and 1.5 GHz in the forward and backward sweeps. This result indicates that the microwave transmittance is modulated as a function of past inputs, and it comes from the hysteresis of MD modulation processes.

## Discussion

Our experimental results show that the response of present device changed with non-volatile memory effect as a function of frequency, power, incident direction and history of the input microwave signal. Therefore, the present device can be regarded as a photonic (or wave) memristive device that can be tuned in a reversible and a non-volatile manner by the wave frequency and amplitude. In electronics, memristive systems are implemented by circuit elements having a memory effect, such as memristors, memcapacitors and meminductors, and a generalized expression on the response of those elements is given as[10,11]:

$$y = g(x, u, t)u, \qquad (6)$$

$$\frac{dx}{dt} = f(x, u, t), \qquad (7)$$

where, $u$ and $y$ denote the input and output of the system, and $x$ denote the state of the system. As the response of microwave device is usually described in terms of scattering parameter

($S$-parameters)[19], one can extend the equations (6) and (7) for the memristive microwave system that its state changes as a function of amplitude and frequency of an incident microwave as:

$$b = S(Z_s)a, \qquad (8)$$

$$\frac{dZ_s(\omega, a_0)}{d\tau} = f(Z_s, \omega, a_0), \qquad (9)$$

$$a = a_0(\tau)\exp j[\omega(\tau)t + \varphi], \qquad (10)$$

where, $S$ is the scattering parameter describing the microwave response of the system under test, $a$ and $b$ are the incident and reflected/transmitted microwaves, $Z_s$ is a variable impedance that describes the microwave mempedance of the memristive microwave system, $a_0$, $\omega$ and $\varphi$ are the amplitude, frequency, and phase of incident microwave, modulated in the timescale $\tau$. It is important to distinguish the timescales of the oscillation of the microwave ($t$) and that of the frequency and amplitude modulation ($\tau$). Unlike analog systems operated by a d.c. or a low frequency a.c. signal, in which the information is encoded in the amplitude of a transient current or voltage, analog wave systems encode information in a slowly varying amplitude and frequency of a carrier wave[35–38]. Therefore, the state variable should be defined in the timescale of the wave modulation, and a transient response change that occurs within a period of wave oscillation may have no proper meaning in the memristive wave information processing. In this light, the term microwave mempedance (or wave-mempedance) used in present study should be distinguished from the mempedance[39,40] used in studies on the memristive microwave devices[9,39–42] that apply the conventional memristive elements because the internal state should be a function of frequency and amplitude of the input microwave signal. Because the electric current or voltage of a wave signal oscillate in time, internal state changes of conventional memristive elements, such as the electrical conductivity in the memristor, will be cancelled out in a period of wave oscillation. In such case, if there is no contribution of second order terms of the current or voltage to the internal state change, the mempedance may not be a well defined quantity because it will depend on a last transient current or voltage change of the carrier microwave signal that has no practical meaning in the information processing.

The $S$-parameters and $Z_s$ can be found provided a proper circuitry that describes the microwave device. For present device, the device response changes as a function of magnetic susceptibility of the YIG, and the N-FMR frequency that determines the magnetic susceptibility changes as a function of modulation microwave frequency and power. Then, the memristive expression of present device for the transmitted microwave is as follows (Supplementary Note 1):

$$b_t = S_{21}(Z_s)a \approx \left(1 - \frac{Z_s}{2Z_0}\right)a \equiv g(x, u, t)u, \qquad (11)$$

$$\frac{dZ_s}{d\tau} = -jC\frac{d\chi}{d\omega_r}\frac{d\omega_r(a_m, \omega_m)}{d\tau} \equiv f(x, u, t), \qquad (12)$$

where, $a$ and $b_t$ is the incident and transmitted microwave, $j^2 = -1$, $C$ is a constant, $\omega_r$ is the N-FMR frequency, $a_m$ and $\omega_m$ are frequency and power of the modulation microwave. Here, we focus on the N-FMR frequency change, and a detailed expression on the magnetic susceptibility change is presented in Supplementary Equation 15 in Supplementary Note 1, The change of N-FMR frequency comes from a variation of MD structure caused by a strong magnetic precession of the MDs. For a simple case that the modulation microwave populate MDs having a N-FMR frequency of $\omega_{r,m}$, one can express the N-FMR

frequency of modulated MD structure as below:

$$\omega_{\mathrm{r}}(\tau) = \omega_{\mathrm{r,i}}(1 - p_{\mathrm{m}}(\tau)) + \omega_{\mathrm{r,m}}p_{\mathrm{m}}(\tau), \qquad (13)$$

where, $\omega_{\mathrm{r}}$ is the total effective N-FMR frequency of the MD structure, $\omega_{\mathrm{r,i}}$ is the N-FMR frequency of an initial MD structure, $p_{\mathrm{m}}$ is the population of MDs having the N-FMR frequency of $\omega_{\mathrm{r,m}}$. Then, by assuming that the population of MDs is proportional to the amplitude of modulation microwave, one can find the dynamic equation of the N-FMR frequency change as below:

$$\frac{\mathrm{d}\omega_{\mathrm{r}}}{\mathrm{d}\tau} = \frac{-(\omega_{\mathrm{r,i}} - \omega_{\mathrm{r,m}})}{a_{\mathrm{sat}}}\frac{\mathrm{d}a_{\mathrm{m}}}{\mathrm{d}\tau}, \qquad (14)$$

where, $a_{\mathrm{sat}}$ is a amplitude of the modulation microwave to saturate the MD population, $a_{\mathrm{m}}$ is the amplitude of the modulation microwave. The solution of equation (14) has a form of $\omega_{\mathrm{r,m}} + (\omega_{\mathrm{r,i}} - \omega_{\mathrm{r,m}})\exp[-\int \mathrm{d}a_{\mathrm{m}}/a_{\mathrm{sat}}]$, showing that the N-FMR frequency depends on the history of applied modulation microwave power.

In conclusion, we report the non-volatile microwave impedance memory effect in YIG that is reversibly tuned by varying the microwave frequency and power. Our experimental results show that the MDs are modulated as a function of frequency, power, and the near field structure of the modulation microwave frequency, and the change of MD structure results in an enhancement of the N-FMR effect around the modulation microwave frequency that is responsible for the impedance memory effect. This can be explained by the MEPP that strongly driven MDs by the modulation microwave evolve to have a structure that can more store and dissipate the precession energy, where the system's entropy production increased along with the increase of energy dissipation that is maximized at the resonance condition. Utilizing the wave frequency, amplitude, and near-field structure to the memory operation might open up new direction in the realization of wave-based adaptive computations, where the wave impedance memory device might be used as a building block of the system.

## Methods

**Microwave response modulation from arbitrary initial states.** A single crystal YIG thin film (thickness ∼1 μm) with in-plane magnetic anisotropy, which was grown on a gadolinium gallium garnet (111) substrate by the liquid phase epitaxy technique, was used for the experiment. The YIG surface was coated by bismuth substituted yttrium iron garnet thin film (∼20 nm) by metal organic decomposition method[43] to enhance the magnetic optical effect[43]. The GGG/YIG/Bi-YIG sample was placed on the HIL of the SILPF, where the substrate layer (GGG) was faced to the SILPF (Supplementary Fig. 12c). The Bi-YIG surface was monitored by transverse MO Kerr measurement system, and at the same time, scattering parameters ($S$-parameter) of the device was monitored by the network analyser (Supplementary Fig. 12b). Before each modulation, initial magnetic domain (MD) structures were prepared by applying a static magnetic field parallel to the LO-axis for a second, and subjected by the modulation microwave after initial MO and scattering parameter ($S$-parameter) measurement. The MO and $S$-parameter measurements for modulated states were conducted after the modulation with few seconds of sleep time, where the sleep time was introduced to reduce a device's response change caused by temperature change. The frequency and power of modulation microwave were swept in forward (increase) and backward (decrease) directions continuously.

**Reversible microwave response modulation.** The sample and experimental methods were the same to that of measurement for arbitrary initial states except for that the measurements were conducted with a fixed initial MD structure. The frequency and power of modulation microwave were swept in forward (increase) and backward (decrease) directions continuously for a given initial MD structure.

**Transverse magneto-optical Kerr effect measurements.** Supplementary Fig. 12c illustrates the experimental setup for transverse magneto-optical Kerr effect (TR-MO) imaging. The YIG thin film was coated by a thin bismuth-substituted yttrium iron garnet (Bi-YIG) thin film (50 nm) by the metal organic decomposition method[5]. The Bi-YIG layer was introduced to enhance the TR-MO

effect of the sample because that there was no observable TR-MO signal for the bare YIG sample. The Bi-YIG/YIG sample was place on the SILPF, where the substrate layer (GGG) was faced to the SILPF. The measurement was conducted in the transverse configuration: the polarization direction of the probing light was perpendicular to the TR-axis, where the angle ($\theta$) between the incident and reflected light was 50–60 degree. For the probing light, blue-wavelength (∼430 nm) LED light source was used because the MO-TR effect was strong around that wavelength. Before the main measurements, two background images were measured under external static magnetic field (∼15 mT) whose polarization direction was parallel (anti-parallel) to the TR-axis. The MD images were calculated by following equation:

$$I_{\mathrm{cal}} = \pm \frac{2I(0) - (I(H) + I(-H))}{I(+H) - I(-H)} \propto M_{\mathrm{TR}}, \qquad (15)$$

where $I_{\mathrm{cal}}$ is the calculated intensity, $I(+H)$, $I(-H)$ are the two background intensity measured under static magnetic field parallel ($+H$) and anti-parallel ($-H$) to the TR-axis, and the sign was chosen so the calculated intensity is positive for the MD whose moment was parallel to the TR-axis. For each measurement, 100-frames were captured by the CCD camera, the measured images were averaged before MD image calculations.

**Microwave transmittance measurement for the SILPF/YIG/GGG-configuration.** A bare YIG thin film that covers three HILs of the SILPF was used for the experiment. The YIG sample was placed on the SILPF, where the YIG layer was faced to the SILPF to enhance the interaction between the H-MWNF and magnetic domain of the YIG (Supplementary Fig. 12d,e). The microwave signals for the probing and modulation process were applied by a microwave sweeper, and the transmitted power was measured by a microwave spectrometer (Supplementary Fig. 12a). The transmittance and MO measurement were conducted with a fixed initial MD structure, and the frequency and power of the modulation microwave were swept continuously. The measurements for MO and microwave transmittance were conducted at the same time, while the magnetic microwave near field imaging was conducted separately. The flowchart of measurement procedures is shown in Supplementary Fig. 13.

**Magneto-optical measurements for the SILPF/YIG/GGG-configuration.** When the YIG layer was faced to the SILPF, the TR-MO measurement was impossible because of a strong absorption of the YIG layer for the blue-wavelength light that prevents measurement of the reflected light from the Bi-YIG layer. To visualize MD structure of the YIG that was faced to the SILPF, we changed the light source to green-wavelength (530 nm) LED because the YIG material is transparent to this frequency. In addition, a bare YIG thin film without Bi-YIG coating was used for the experiments because that the Bi-YIG-coated sample showed no measurable MO signal related to the Bi-YIG layer. The incident angles of the light beam, $\theta$ and $\phi$, and the angles of polarizer and analyser were carefully modulated, and a strong optical signal related to the magnetic domains of the YIG appeared when the CCD was slightly deviated from the propagation direction of the reflected light as illustrated in Supplementary Fig. 12d,e. We tested the dependency of the MO signal on the magnetization direction and the polarization direction of the incident light by changing applied static magnetic field, polarizer and, analyser angle. It was observed that there was no dependency of the MO signal on the applied magnetic field direction and polarization direction of the incident light. In particular, the MO signal only appeared under no (or a weak) external static magnetic field. Because the MO signal did not depend on the magnetization direction, it may be related to a quadric MO effect[44] with a complicated MO process that depends on a magnetic domain structure of the YIG. Although the MO signal in the present measurement method did not give quantitative information on the magnetization state, it showed a strong change depending on the MD structure of the YIG. Supplementary Fig. 14a shows MO images measured under static external magnetic field, and Supplementary Fig. 14b,c show MO images measured under no external magnetic field, and measured after the modulation microwave sweeps. While the there was no contrast of the optical signal when the external magnetic field was applied, the MO contrast appeared when the external magnetic field was removed, and it showed a strong change around the HIL of the SILPF after the modulation microwave sweeps. This result indicates that the present MO signal can be utilized to characterize the change of MD structure, and further detailed study on this MO signal will be reported elsewhere.

**Magnetic microwave near-field imaging.** The magnetic microwave near-field distributions were visualized by calculating a heat source distribution caused by the microwave Eddy current of a conducting thin film deposited on a glass substrate. When the microwave is applied on the device under test, the glass substrate is heated by the Eddy current induced in the metal thin film as a function of magnetic microwave near field distribution. The generated heat cause a thermal stress of the glass substrate, and the stress distribution can be measured by photoelastic measurement (Supplementary Fig. 15a). By determining the heat source distribution through a plane thermal stress analysis, the magnetic microwave near field can be visualized. Details on the measurement process are presented in Supplementary Fig. 13b. This method is suitable for a high resolution mapping of magnetic

microwave near-field under zero-field ferromagnetic resonance condition because a biasing magnetic field, which is commonly needed in existing magnetic microwave near field mapping systems, is not required for the measurement.

Photoelastic measurements were conducted by the polarization modulation technique[45]. Supplementary Fig. 15b shows a schematic diagram of our measurement system, where a green light emitting diode (LED, $\lambda \approx 530$ nm) was used as the light source, and two sheet-type polarizers were used as the polarizer and analyser. The polarization of incident light was modulated by the liquid crystal modulator (LCM) to be in the left-handed or right handed circular polarized state. The LCM acts as a variable linear retarder, whose retardation is modulated by an a.c. electric field so that the polarization state of the input beam can be modulated by changing a driving voltage without mechanical rotation. The continuous wave (CW) microwave was applied to the device under test (DUT) by a HP-83620 sweeper, and the transmitted power was monitored by a spectrometer. The changes of light intensity were monitored by the CCD camera, where two measurements were conducted when the analyser was aligned to 90 and 45 degrees (Supplementary Fig. 15c). Operations of all devices and data analysis were conducted by a computer with a custom program. Detailed information about the calculation of the heat source distribution from the photoelastic measurement results is presented in Supplementary Notes 6 and 7.

**Calculation of N-FMR frequency.** The N-FMR frequencies of modulated states were calculated by fitting vertical line-profiles of the $\Delta S_{21,p}$ to the Gaussian line-shape functions as:

$$y = a \exp\left[ -\left( \frac{f_p - f_{peak}}{c} \right)^2 \right], \tag{16}$$

$$y = a \left( \frac{f_p - f_{peak}}{c} \right) \exp\left[ -\left( \frac{f_p - f_{peak}}{c} \right)^2 \right], \tag{17}$$

where the $f_p$ is probing microwave frequency, $f_{peak}$ is the N-FMR frequency to be calculated, $a$ and $c$ are constants related to the amplitude and line-width of the $\Delta S_{21,p}$. The real and imaginary parts of the part of $\Delta S_{21,p}$ were fitted by equations (16) and (17), respectively, where an additional linear function was included to the equations to compensate an overall decrease of the curves. Supplementary Fig. 17 shows representative example of the fitting results for real (a) and imaginary (b) parts of the $\Delta S_{21,p}$. Deviations between the fitting and measurement curves may result from oversimplified assumptions of the L-model[23], and from a contribution of N-FMR effect of pre-existing MDs. Because our interest is the N-FMR frequency shift rather than its exact value, we used the above equations with the oversimplified assumptions.

**Data availability.** The data that support the findings of this study are available from the corresponding author on request.

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

## Acknowledgements

This work was supported by Basic Science Research Program through the National Research Foundation of Korea (NRF) funded by the Ministry of Education (2015R1D1A1A02061824)&(2009-0093822).

## Author contributions

H.L. conceived and conducted all experiments and analysed the data, H.L., B.F. and K.L. contributed basic idea and co-wrote the manuscript. All authors discussed the results and commented on the manuscript.

## Additional information

**Competing financial interests:** The authors declare no competing financial interests.

