## [Peer Review File · Nature Communications]

Reviewers' comments:

Reviewer #1 (Remarks to the Author):

The authors presented an interesting study that exploits the possibility of modulating magnetic domains via H-MWNF, presenting analogies to ReRAM and memristive systems memory state modulation capacities.

The main concern with this work is that this resemblance is only supported qualitatively, with no direct evidence of the physics governing the interaction of the H-MWNF with the supporting materials. What are the specific processes involved here? The authors have very nicely and extensively commented on the resulting effects of such changes, i.e. impact on S-parameters. But how are the EM properties of the materials affected to support these phenomena?

Similarly, what are the governing material implications behind the reversible and irreversible processes described therein, both for setting and resetting the system's memory? Clearly this is a fundamentally different mechanism from ReRAM, and the authors have nicely argued the impact of different fm due to the spatial distribution of H-MWNF; the underlying mechanisms need however mentioning for completing the study.

The term mempendance has been previously used in a few occasions for describing the modulation of electronic states of devices, whether resistance, capacitance or inductance. It would be useful to clarify any resemblance or not with prior art in the field.

As authors discuss the resemblance with memristive systems, I encourage them to add onto Fig. 1 a schematic that demonstrates the equivalent thresholds above which MD modulation is possible.

It would be very useful for the authors to demonstrate what happens when continuous MW is used at fm's that are below 1GHz or above 2GHz, when no significant MD modulation occurs. And it would also be beneficial for the authors to demonstrate whether MD modulations are volatile or non-volatile.

In p. 5, the authors discuss the impact of locally intensive H-MWNF and the resulting MD instabilities observed. Can the authors suggest any possible design and/or bias control parameters for mitigating this challenge?

The authors also identify the MD state dependence (history dependence) impact, for example in p. 6. In this case, this is described only for within the usable fm region that causes MD modulations. What happens however outside this space? And particularly, what happens during the 1st "MW programming event" that eventually renders history dependence on subsequent stimuli?

Minor comment: Fig. 1d needs to be depicted in a clearer way; cannot discriminate which cycle is which.

Reviewer #2 (Remarks to the Author):

In this paper, Lee et al reported demonstration of a nonvolatile adaptive microwave impedance memory effect in YIG and showed that the magnetic domain structure in the ferromagnet can be reversibly modulated by regulating the input microwave frequency, utilizing a near-field structure.

Correlation between magnetic domain controls and microwave modulation was established. This paper is interesting and timely. However, the following comments need to be addressed before the manuscript can be further considered.

1. It appears the authors presented a lot of results but failed to present a systematic explanation (and ideally a quantitative explanation). For example, as shown in Figure 1d, the transmitted power can be reversibly decreased and increased in a microwave frequency range between 0.5 GHz and 1.0 GHz during forward and backward sweep process. Why does the transmitted power increase and decrease abruptly at a frequency of 1.0 GHz and respectively? In addition, it is obviously that at around 0.75 GHz, a peak was always observed during the repeated sweeping process. Please explain this phenomenon.
2. Although the authors have extensively studied the evolution of the transmitted power during modulation process, the detailed information about the magnetic domain property changes, such as magnetic moment, magnetization and domain wall movement have not been well analyzed, which can provide more useful information about the microwave controls.
3. The authors should briefly discuss why YIG material is used in this study.
4. The authors claimed that the local magnetic domain structure can affect the transmittance of the microwave and showed related calculation results of the correlation between the magneto optical signal and transmitted signal. The comparison of the modulation results between YIG with different magnetic structures should then be performed to further exam the calculation results.
5. The authors mentioned that the magnetic domain structure showed a local instability, which is attributed to the strong spin precession from the intensive H-MWNF. Will this fluctuation be suppressed by decreasing the power of the excitation microwave, and thus reduce the variation of domain structures during repeated modulating cycles?
6. One of the characteristics of analog memristor is the gradual switch of the output signal by the number of input pulse signal. Can the transmitted power be gradually controlled by controlling the number of the input excitation microwave pulse signal?
7. Finally, the "state variables" in Eq. (2) need to be clearly identified, along with the specific form of equation (3) that describe the dynamics of the state variable(s). Identifying the state variable(s) is a key component in the memristor theory framework and the authors cannot simply gloss over this point by stating that "N is a set of internal states of the device" and "The h (function) denoting the internal state change of the device is a multivariable function of modulation MW frequency, power, and direction" without any further elaboration.

REVIEWERS' COMMENTS:

Reviewer #1 (Remarks to the Author):

The authors have substantially amended their manuscript, improving the quality to the level that this is now acceptable for publication in Nature Communications. All my previous concerns have been addressed. The authors have now presented a lot of new experimental data that allow the reader to quantify the phenomena observed.

Reviewer #2 (Remarks to the Author):

The authors have performed extensive new experiments and addressed my comments satisfactorily. The manuscript can now be accepted for publication.

Comments to the editors and reviewers

About the experiment methods

The main concern of the reviewers on our previous manuscript was the lack of a quantitative explanation of the experimental observations. Experimental results in the previous manuscript do not have enough information to address the reviewer's concerns, because the quantitative measurement of magnetic susceptibility and magnetic domains changes is required for the quantitative explanations. Therefore, we modified experimental methods for the microwave response measurement and magnetic domain imaging as below:

1. Microwave response measurement

In the previous manuscript, we only discussed the change of transmitted microwave power as a function of modulation microwave, where the measurements were conducted by measuring the transmitted power by a microwave spectrometer. Because the magnetic susceptibility (or permeability) is a complex quantity, which is responsible for the impedance change of the device, information on the correlated phase changes between the incident and transmitted microwave is also needed to identify the magnetic susceptibility changes. Therefore, we modified the experimental method as below: **the real and imaginary parts of the magnetic susceptibility change were calculated from the scattering parameter measurement results in a quantitative way.**

2. Magnetic domain imaging

Our previous magneto-optical imaging method only gave information of a structural change of the magnetic domains, and thus, we discussed qualitatively for the changes of magnetic domain structure and device response in the previous manuscript. Because the bare YIG film has a weak magneto optical (transversal and longitudinal) Kerr effect, we modified the YIG sample by coating a Bi-YIG material having a high magneto optical activity. The surface of Bi-YIG was monitored by a well defined transversal magneto optical Kerr effect measurement system. From the measurement results, **we calculated the magnetization direction of the magnetic domains quantitatively, and identified the changes of magnetization direction and domain wall shift caused by the modulation process.**

About the manuscript revision

Many parts of the previous manuscript were revised with new experimental results. We present brief explanations on each section of the present manuscript.

1. Introduction

2. Results

Operation principle of the natural ferromagnetic resonance (N-FMR) device: We describe how the N-FMR effect of the YIG changes the response of the device. We presented relations between the N-FMR frequency, magnetic domain structure, and the magnetic susceptibility of the YIG, and explained the device response change from the relations.

Microwave response modulation from arbitrary initial states: We investigated an overall tendency of the device's response change caused by the modulation process. We presented experimental results on the change of magnetic susceptibility and magnetic domain structure as a function of modulation microwave frequency and power with various initial magnetic domain structures. Based on the results, we discussed how the modulation microwave modified the magnetic domains, and how the modulated magnetic domains resulted in a change of device response.

Reversible microwave response modulation: We investigated reversible change of device's response by the modulation process. We presented experimental results on the changes of magnetic susceptibility and magnetic domains caused by a continuous modulation process for a given initial state. We discussed how the modulation microwave resulted in a reversible change of magnetic domain structure and device's response.

Microwave response modulation by H-MWNF structure: We investigated a contribution of H-MWNF structure to the device's response. We presented experimental results appearing in the previous manuscript fig. 2, and discussed how the magnetic domain structure and device's response change based on new experimental results appearing in above sections.

3. Discussion

In this section, we discussed the resemblance and difference of the term 'mempedance' used in present study with prior art in the field. We presented a more well defined state variable and state equation describing present device for a simple case.

4. Supplement text and figures

Supplement text

S1. Relations between the magnetic susceptibility, N-FMR frequency, and device response are presented with a simple practical example for the Landau flux-closure magnetic domain structure.

S2.1. Explanations on the transversal magneto optical Kerr measurement are presented.

S2.2. Explanations on the magneto optical measurement method used in previous manuscript are presented (Equivalent to S1 in previous manuscript).

S2.3. Explanations on the scattering parameter measurement method are presented.

S2.4. Explanations on the microwave transmittance measurement method used in previous manuscript are presented.

S3. Equivalent to S2 in previous manuscript.

S4. Explanations on the calculation of N-FMR frequency from scattering parameter measurement results are presented.

S5. Descriptions on the magnetic domain structure and device's response changes as a function of modulation microwave power are presented.

S6. Equivalent to S4 in previous manuscript.

S7. Equivalent to S3 in previous manuscript.

S8. Equivalent to descriptions of previous experimental results appeared in figure 3 in previous manuscript.

S9. Reference: Additional references (7 items) are presented for supplement text S1, S2.1, and S4.

Responses to the Reviewer's comments:

Reviewer #1

The authors presented an interesting study that exploits the possibility of modulating magnetic domains via H-MWNF, presenting analogies to ReRAM and memristive systems memory state modulation capacities.

1. The main concern with this work is that this resemblance is only supported qualitatively.

atively, with no direct evidence of the physics governing the interaction of the H-MWNF with the supporting materials. What are the specific processes involved here? The authors have very nicely and extensively commented on the resulting effects of such changes, i.e. impact on S-parameters. But how are the EM properties of the materials affected to support these phenomena?

The main physical mechanism involved in the present study is the natural ferromagnetic resonance (N-FMR) effect of the magnetic material. The N-FMR effect occurs when the applied microwave frequency is close to the natural frequency (or Larmor frequency) of the magnetic precession. The N-FMR frequency depends on the internal effective field acting on the magnetic domain in the ferromagnetic material, and the internal effective field is a function of demagnetization field of the magnetic domain. Because the demagnetization factor determining the demagnetization field depends on the geometrical shape of the magnetic domains, the N-FMR frequency changes along with a variation of magnetic domain structure, and the change of the N-FMR frequency results in a change of dynamic magnetic susceptibility. In the present device, the response is determined by the dynamic magnetic susceptibility of the coupled magnetic material, and as a result, the device response changes along with a variation of the magnetic domain structure. Therefore, the dynamic magnetic susceptibility is the responsible EM property for the device response change, and the governing physics is the resonant magnetic precession of magnetic domain caused by the H-MWNF. We present experimental results on the magnetic susceptibility changes in our revised manuscripts (second and third subsections in the Results section) that can be a direct evidence of our explanations, and presented detailed descriptions relations between the magnetic susceptibility, magnetic domain structure, and N-FMR effect in the supplement text S1 with a simple quantitative physical model.

2. Similarly, what are the governing material implications behind the reversible and irreversible processes described therein, both for setting and resetting the system's memory? Clearly this is a fundamentally different mechanism from ReRAM, and the authors have nicely argued the impact of different fm due to the spatial distribution of H-MWNF; the underlying mechanisms need however mentioning for completing the study.

The reversibility of the present device comes from the thermodynamic direction of strongly driven ferromagnetic system that the magnetic domains evolve to maximize their precession

energy according to the maximum entropy production principle (MEPP) as discussed in reference #20 (Krasnyuk, A. *et al. Phys. Rev. Lett.* **95**, 207201 (2005)) of the present manuscript. When the precession energy of magnetic domains exceeds some critical value, the magnetic domains start to change resulting in a variation of N-FMR frequency. When the variation of N-FMR frequency of a driven magnetic domain results in an increase of its precession energy, in which it occurs when the N-FMR frequency of the magnetic domain shifts to the driving microwave frequency, the magnetic domain start to grow by the increase of precession amplitude. The growth of magnetic domain occurs until the precession energy gain by the N-FMR frequency shift is vanished, and the grown domain can remain stable even in an absence of applied microwave provided that there is no internal torque acting on the magnetic domain. As a result, the modulated magnetic domains show an enhancement of N-FMR effect resulting in a non-volatile change of device response (setting system's memory). Because the precession energy is maximized around the resonance condition, the N-FMR frequency of the modulated magnetic domains shifts to the driving microwave frequency. The N-FMR frequency can increase or decrease depending on the driving microwave frequency (setting the system's memory), and thus, the reversibility of the device response is achieved. Therefore, the underlying mechanism is the non-linear process of the strongly driven magnetic domains following the maximum entropy production principle. We presented detailed descriptions on the mechanism of the reversible modulation of magnetic domain and device's response in present manuscript (third subsections in the Results section) with experimental results supporting our discussion.

3. The term mempedance has been previously used in a few occasions for describing the modulation of electronic states of devices, whether resistance, capacitance or inductance. It would be useful to clarify any resemblance or not with prior art in the field.

We presented detailed descriptions regarding the resemblance and difference of the term 'mempedance' used in present study in the Discussion section.

4. As authors discuss the resemblance with memristive systems, I encourage them to add onto Fig. 1 a schematic that demonstrates the equivalent thresholds above which MD modulation is possible.

We revised the illustration on the device operation, where the representative measurement results appeared in Fig. 1d was omitted. In the Fig. 1, we illustrated the operation principle of

the device in more detail, and presented an illustration showing a threshold behavior of the magnetic domain modulation process in Fig. 1f. In addition, we presented experimental results on the device's response change as a function of modulation microwave power in Fig 2.f and supplement text S5 with a description on the threshold behavior of the magnetic domain modulation.

5. It would be very useful for the authors to demonstrate what happens when continuous MW is used at fm's that are below 1GHz or above 2GHz, when no significant MD modulation occurs. And it would also be beneficial for the authors to demonstrate whether MD modulations are volatile or non-volatile.

We presented experimental results on the modulation process for an arbitrary and for a given initial condition in second and third subsections of the Results section, respectively. In these sections, we discussed the frequency dependency of the modulation process in all frequency ranges. We used the term 'non-volatile' to mean a memory state that is stable in an absent of applied microwave signal and external magnetic field, and is not changed until the subsequent modulation process is applied. About the non-volatile memory effect, we described it in the operation principle section of present manuscript: 'If the modified MD structure is stable so that there is no internal torque acting on its magnetic moment, it will remain even when the microwave signal is stopped (modulation process). Then, the device response will show a non-volatile change from a variation of the N-FMR frequency of the modified MD structure.' As our experiment is conducted with a 'sleep time' between the modulation and probing processes, in which there was no external microwave and static magnetic field during the sleep time (detailed measurement process was presented in supplement figure S3a), the measurement results on the device response and magnetic domain describe non-volatile changes.

6. In p. 5, the authors discuss the impact of locally intensive H-MWNF and the resulting MD instabilities observed. Can the authors suggest any possible design and/or bias control parameters for mitigating this challenge?

Because the fluctuation of magnetic domain structure occurs due to their strong magnetic precession, a decrease of applied modulation microwave power results in a decrease of the fluctuation. As shown by the new experimental results, the magnetic domains showed well defined reversible change by the modulation microwave with reduced magnetic domain fluctuation when the amplitude of the H-MWNF was reduced. We expect a more stable MD

modulation can be realized by a proper designing of the H-MWNF structure, or by changing magnetic property of material. Because the magnetic property depends on various parameters such as stress, magneto-crystalline anisotropy, thickness, temperature, external magnetic field, and so on, more extensive studies will be required, and we postpone it for our upcoming work.

7. The authors also identify the MD state dependence (history dependence) impact, for example in p. 6. In this case, this is described only for within the usable f_m region that causes MD modulations. What happens however outside this space? And particularly, what happens during the 1st "MW programming event" that eventually renders history dependence on subsequent stimuli?

We presented experimental results and descriptions on the initial modulation process in continuous modulation sweep in third subsection of the Result sections, and supplement text S5 and S7. In addition, we also discussed about the frequency dependency of the modulation process in the second and third subsection of the Result section. We summarize the discussions as below:

Modulation outside of usable f_m region: The modulation process is driven by the precession energy of the magnetic domains, and the precession energy is a function of frequency and amplitude of the driving microwave. In addition, the precession energy strongly depends on the resonance frequency of the magnetic precession, where it increases rapidly as the driving microwave frequency approaches the resonance frequency. For a case that f_m is far from the resonance frequency (outside of usable f_m region), the precession energy is small compared to that of f_m close to the resonance frequency (usable f_m region). Therefore, a more intense microwave is required for the f_m outside of usable f_m region to modulate the magnetic domains compared to that of f_m usable f_m region, and the usable f_m region is determined by the driving microwave intensity. Outside the usable f_m region, the magnetic domain precess by the microwave but do not change their structure because that the precession energy is not enough to overcome the internal effective magnetic field.

1st MW programming event: During the initial modulation, the magnetic domains start growing when their precession energy exceed some critical value so that the magnetic domain modulation occurs. For a constant power of modulation microwave, the amplitude of magnetic precession increases rapidly as the microwave frequency approaches the resonance frequency. Therefore, the modulation only occurs when the f_m is in the range of usable f_m

region, and out of the region, the magnetic domains just precess without changing their structure. Depending on the initial magnetic domain structure, the magnetic domains grow through a magnetic domain wall shift. The grown magnetic domain during the initial modulation (1st programming event) can be stable for the subsequent modulation microwave because of an increase of internal effective field by the increase of magnetization through the domain growth, where the stability is determined by the amplitude of the driving microwave.

8. Minor comment: Fig. 1d needs to be depicted in a clearer way; cannot discriminate which cycle is which.

We revised the illustration on the device operation, where the representative measurement results appeared in Fig. 1d was omitted. We presented equivalent experimental results in Fig. 4b, where the change of device response presented as a function of sweep index to show more clearly the reversibility of the device response modulation.

Reviewer #2

In this paper, Lee et al reported demonstration of a nonvolatile adaptive microwave impedance memory effect in YIG and showed that the magnetic domain structure in the ferromagnet can be reversibly modulated by regulating the input microwave frequency, utilizing a near-field structure. Correlation between magnetic domain controls and microwave modulation was established. This paper is interesting and timely. However, the following comments need to be addressed before the manuscript can be further considered.

1. It appears the authors presented a lot of results but failed to present a systematic explanation (and ideally a quantitative explanation). For example, as shown in Figure 1d, the transmitted power can be reversibly decreased and increased in a microwave frequency range between 0.5 GHz and 1.0 GHz during forward and backward sweep process. Why does the transmitted power increase and decrease abruptly at a frequency of 1.0 GHz and respectively? In addition, it is obviously that at around 0.75 GHz, a peak was always observed during the repeated sweeping process. Please explain this phenomenon.

We presented more detailed and quantitative explanation on the device response change and magnetic domain modulation with new experimental results. We presented descriptions on the relations between the device response, magnetic domain structure, and magnetic

susceptibility in the operation principle subsection in the Result section, and presented more detailed descriptions in the supplement test S1 with a simple practical example. The reviewer can find summarized the descriptions on the physical mechanism of the response modulation in the responses #1 and #2 for the reviewer #1. About the decrease and increase of transmitted microwave power, we respond as follow: The decrease of transmitted microwave power comes from the increase of resonant absorption by the magnetic precession. The transmitted microwave power can increase or decrease depending on the resonance frequency, where the transmitted power is minimized when the probing microwave frequency is identical to the N-FMR frequency, and transmitted power increases as the probing microwave frequency gets away from the N-FMR frequency. As presented in the new manuscript and responses #1,2 for the reviewer #1, the modulation microwave modifies the magnetic domain to have a resonance frequency close to the modulation microwave frequency. Therefore, the device response changes depending on the modulation microwave frequency and the change is determined by the N-FMR frequency of the modulated magnetic domain structure.

2. Although the authors have extensively studied the evolution of the transmitted power during modulation process, the detailed information about the magnetic domain property changes, such as magnetic moment, magnetization and domain wall movement have not been well analyzed, which can provide more useful information about the microwave controls.

We couldn't identify such detailed information from the previous MO imaging results (details on the previous MO imaging method and results are presented in supplement text S2.2). We present new experimental results on the magnetic domain structure with modified experimental method (presented in supplement text S2.1). We used a quantitative transversal magneto optical Kerr effect measurement system, and presented quantitative information on the magnetization direction, domain wall movement, and so on.

3. The authors should briefly discuss why YIG material is used in this study.

We describe the importance of YIG material briefly with a new reference: 'where the YIG material has a particular importance due to its outstanding properties such as a high quality factor (Q-factor) with a narrow FMR line-width and extremely low loss'¹⁷, (¹⁷ Serga, A. et al. YIG magnonics. *J. Phys. D: Appl. Phys.* **43**, 264002 (2010))

4. The authors claimed that the local magnetic domain structure can affect the transmittance of the microwave and showed related calculation results of the correlation between the magneto optical signal and transmitted signal. The comparison of the modulation results between YIG with different magnetic structures should then be performed to further examine the calculation results.

In the previous manuscript, we presented the correlation distribution to support qualitatively that there is a relation between the device response modulation and modulated magnetic domain structure. This has done because our previous MO imaging does not give information on the magnetization, domain structure, and domain wall shift, and so on. As our new experimental results showed well defined magnetic domain structure and their change, and as we presented a quantitative explanation on the changes of magnetic susceptibility and device response, we omitted the correlation distribution results.

5. The authors mentioned that the magnetic domain structure showed a local instability, which is attributed to the strong spin precession from the intensive H-MWNF. Will this fluctuation be suppressed by decreasing the power of the excitation microwave, and thus reduce the variation of domain structures during repeated modulating cycles?

The strong fluctuation of magnetic domain structure occurred when the YIG was faced to the SILPF, and when the applied microwave power was strong. Because the fluctuation of magnetic domain structure occurs due to their strong magnetic precession, a decrease of applied modulation microwave power results in a decrease of the fluctuation. As shown in new experimental results, where the substrate layer of the YIG thin film was faced to the SILPF so that the interaction between the YIG and H-MWNF decreased, the magnetic domain showed well defined reversible change by the modulation microwave. This result well demonstrates our suggestion that the decrease of microwave power reduce the fluctuation of the magnetic domain modulation.

6. One of the characteristics of analog memristor is the gradual switch of the output signal by the number of input pulse signal. Can the transmitted power be gradually controlled by controlling the number of the input excitation microwave pulse signal?

We presented results on the response modulation with a continuous sweep of frequency (f_m) and power (p_m) of the modulation microwave. A pulse of modulation microwave (~100ms) was applied with a constant f_m and p_m , and the f_m and p_m were swept continuously. The response of the device changed gradually along with an increase and decrease of f_m and p_m . In

addition, we tested whether the response changed gradually by a series of identical modulation microwaves. We tested various cases by changing the modulation microwave power, the applied time from 1ms to 100ms, and applied frequency. We observed that the applied numbers of identical modulation microwaves didn't give a significant influence on the device response change, where the variation was in the range of noise level. About the results, we suggest that the modulation process is finished less than 1ms, in which it is a minimum time scale of the modulation microwave pulse generated by our instrument. Because limitation of our instrument, we couldn't confirm the gradual response change by a series of identical modulation microwaves, and further detailed study is required to address the present issue. However, we note that the device's response can be controlled gradually by changing the f_m and p_m as presented in our manuscript and supplement figures S8c and S12.

7. Finally, the "state variables" in Eq. (2) need to be clearly identified, along with the specific form of equation (3) that describe the dynamics of the state variable(s). Identifying the state variable(s) is a key component in the memristor theory framework and the authors cannot simply gloss over this point by stating that "N is a set of internal states of the device" and "The h (function) denoting the internal state change of the device is a multivariable function of modulation MW frequency, power, and direction" without any further elaboration.

We presented a more well-defined description on the state variable and equations in the discussion section of the present manuscript.